# Tribological Characteristics of a-C:H:Si and a-C:H:SiO_x_ Coatings Tested in Simulated Body Fluid and Protein Environment

**DOI:** 10.3390/ma15062082

**Published:** 2022-03-11

**Authors:** Anna Jedrzejczak, Witold Szymanski, Lukasz Kolodziejczyk, Anna Sobczyk-Guzenda, Witold Kaczorowski, Jacek Grabarczyk, Piotr Niedzielski, Agnieszka Kolodziejczyk, Damian Batory

**Affiliations:** 1Institute of Materials Science and Engineering, 1/15 Stefanowskiego St., 90-924 Lodz, Poland; anna.jedrzejczak@p.lodz.pl (A.J.); witold.szymanski@p.lodz.pl (W.S.); lukasz.kolodziejczyk@p.lodz.pl (L.K.); anna.sobczyk-guzenda@p.lodz.pl (A.S.-G.); witold.kaczorowski@p.lodz.pl (W.K.); jacek.grabarczyk@p.lodz.pl (J.G.); piotr.niedzielski@p.lodz.pl (P.N.); 2Nanomaterial Structural Research Laboratory, Bionanopark Ltd., 114/116 Dubois St., 93-465 Lodz, Poland; a.kolodziejczyk@bionanopark.pl; 3Department of Vehicles and Fundamentals of Machine Design, 1/15 Stefanowskiego St., 90-924 Lodz, Poland

**Keywords:** Si-DLC, wear, coefficient of friction, biomedical engineering

## Abstract

This paper presents the tribological properties of silicon and oxygen incorporated diamond-like carbon coatings tested in simulated body fluid and bovine serum albumin environments. The tests were performed using a ball-on-disc tribometer with an AISI316L steel counterbody. The wear tracks and wear scars were analyzed using optical microscopy and a nanoindenter. The interaction between the coating and the working environment was analyzed by Fourier transform infrared spectroscopy, whereas changes in the chemical structure before and after the tribological tests were compared with the use of Raman spectroscopy. Our study showed that the tribological parameters are governed by the presence of oxygen rather than the changing concentration of silicon. Both of the spectroscopy results confirm this statement, indicating that coatings with low concentrations of silicon and oxygen appear to be better candidates for biological applications in terms of wear resistance.

## 1. Introduction

Wear is a critical issue in the design of medical implants, particularly in load bearing applications such as knee and hip joint prosthesis. Despite noticeable progress in the development of prospective materials including their modification it is still an ongoing scientific and technological challenge. As the rate of failure and the loss of implants are undesirably high it opens a lot of possibilities for improvement. The main factors leading to failure are wear, fatigue, chemical degradation, and infection. Thus, the surface of medical implants must be tolerant to dynamic loading and long-term exposure to biological interaction with surrounding tissue [1,2]. Both may induce severe degradation of the material and the implant as well. Biotribocorrosion processes in the field of biomedical applications include oxidation, degradation of the passive layer, generation and transfer of corrosion products. The corrosion related mechanical wear contributes significantly to material loss and especially to the formation of metal debris and ions [3]. In the case of joint prosthesis, the closed geometry of the contact causes wear debris which is easily trapped and thus introduces an additional type of wear, namely the third body effect. All of which inevitably lead to the excessive wear of the head and the cup of the implant, which in some cases requires surgical re-intervention [4]. Significant quantities of wear products are absorbed by the surrounding tissues, resulting in their migration or accumulation. It is estimated that between 10% and 30% of all endoprosthesis revisions are due to aseptic loosening, mainly caused by wear products formed in the friction node [5,6]. The reduction of biotribological wear can be achieved by producing an optimal top layer dedicated to specific working conditions including temperature, environment, and load. Technological surface layers with increased wear resistance are formed by the appropriate selection of materials for the elements of the joints, characterized by increased hardness, a low coefficient of friction (COF), and good corrosion resistance. One of the most widely studied thin film materials used to enhance implant performance are diamond-like carbon (DLC) coatings. It is well known that DLC films, besides corrosion resistance [7], high hardness [8] and excellent biocompatibility [9,10,11,12], demonstrate low wear and a low friction coefficient (between 0.05 and 0.2) against most of materials [13,14]. On the other hand, high residual stresses and the strong dependence of the friction coefficient on environmental conditions were also registered. Nevertheless, an excellent combination of properties desired particularly in the biomedical engineering field have prompted scientists to look for ways to eliminate them. This has been reflected in the still increasing number of papers concerning the doping of these coatings with other elements, e.g., Si, Ag, Ti, Cu [15,16,17,18,19].

Presently, silicon-incorporated coatings have gathered a great deal of interest. Doping DLC with silicon not only enables the maintenance of high mechanical properties which are typical for DLC coatings, but also allows one to improve other properties such as adhesion, biological, corrosion and tribological properties [15,20,21,22]. This makes them an ideal candidate for protective coatings as used in medicine. Numerous scientific papers report very good tribological properties of newly designed and optimized surface modifications. However, the results obtained from the dry friction tests do not correspond to the real conditions present in the human body. In the design and testing of orthopedic biomaterials, the critical issue is their performance in interaction with body fluids and this requires the use of only clinically relevant synthetic fluids as close substitutes for the real biological environment [23,24,25]. Huang et al. investigated the friction and wear of DLC coatings tested in simulated body fluid (SBF) and human serum albumin (HAS) environments [26]. They proposed a friction mechanism where in aqueous solution an excess of H_2_O molecules cause almost all of the dangling bonds to be terminated by C-H, C-O and C-OOH groups. Therefore, the friction force depends on the hydrogen bonding and Van der Waals force interaction between a tribopair and H_2_O molecules. This may result in a lower value of the COF. Admittedly, the authors registered a much lower value of the COF for the test in SBF. 

The mechanisms of the BSA interaction with the surface of DLC coatings during frictional processes have been discussed in the literature [26,27,28,29,30,31]. They were mainly related to the changes in the protein conformation and formation of hydrogen bonding and Van der Waals interaction between DLC and a counterbody. As the influence of protein on the tribological properties of silicon incorporated carbon coatings has not yet been reported, the present paper aims to fill the gap in this field. In our work we present the effect of silicon and oxygen admixture on the friction and wear behavior of silicon-incorporated DLC coatings tested in SBF and BSA environments. DLC coatings doped with different concentrations of silicon were produced using two silicon precursors. The coefficient of friction in SBF and BSA was determined using a ball-on-disc tribometer. After the tests, the resulting wear tracks on the samples and the wear scars on the counterbodies were analyzed and specific wear rates were calculated. Raman and Fourier transform infrared spectroscopy (FTIR) were used for the chemical characterization of silicon-incorporated DLC coatings before and after the tribological tests. Additionally, tribological tests in ambient air and for bare DLC were performed as a reference.

## 2. Materials and Methods

DLC and silicon-incorporated DLC coatings were deposited on the surface of AISI 316L stainless steel samples 6 mm thick and 16 mm in diameter. Coupons were grinded on various grades of abrasive paper and mirror polished with the use of a diamond suspension. Before the modification process, samples were cleaned ultrasonically in a methanol bath for 10 min and placed in an RF PACVD (Radio Frequency Plasma Assisted Chemical Vapor Deposition) system (TUL, Lodz, Poland) on a water-cooled electrode. The system was pumped to the base pressure of 3 × 10^−3^ Pa. Prior to the deposition, samples were etched in Ar plasma for 10 min under pressure of 2 Pa with a negative self-bias potential of 800 V. Carbon coatings with different concentrations of silicon were deposited using gas mixtures composed of methane (CH_4_), hexamethyldisiloxane (HMDSO, Sigma Aldrich, Saint Louis, MO, USA, NMR grade ≥99.5%) and tetramethylsilane (TMS, Sigma Aldrich, Saint Louis, MO, USA, >99.0% (GC)). Varied concentrations of silicon were obtained by changing CH_4_/HMDSO and CH_4_/TMS gas flow ratio (further denoted as HMDSO1, HMDSO2, TMS1 and TMS2). Given flow ratios were estimated based on the resulting pressure during the deposition in such a way that after decreasing the flow rate of methane, a flow of silicon precursor was increased each time to obtain a stable resulting pressure of 20 Pa. The silicon-incorporated DLC coatings were manufactured under the negative self-bias potential of 800 V which results from our earlier experience in terms of tribological properties of silicon incorporated DLC cooperating with different counterbody materials, whereas the thickness of the deposited coatings in each case was equal to 1 µm (thickness determined according to results from our previous work [32]). In order to ensure a good adhesion of the investigated layers, a gradient Ti–Ti_x_C_y_ interlayer with a thickness of ca. 150 nm was deposited directly before the process of synthesis of the final films. A more detailed description of the synthesis of gradient interlayer as well as the deposition system can be found elsewhere [14]. 

A tribometer in pin-on-disc configuration was used to characterize the tribological properties of the tested surfaces. Although this simple setup does not reproduce all clinical parameters relevant for the accurate simulation of wear (e.g., contact geometry, loading cycle, fluid entrainment or multi-directional sliding), it can give valuable fundamental information about the effects of the interfacial protein adsorption on the friction behavior.

The tribological tests were conducted using the pin-on-disc tribometer (CSM, Graz, Austria) under a load of 1 N, a sliding speed of 0.1 m/s, a sliding radius of 5 mm, within 10,000 revolutions. As the counterbody material is 6.35 mm in diameter, an AISI 316L stainless steel ball was used (G200, Ra–0.2 μm). Taking into account the mechanical properties, especially the Young’s modulus of the tested Si-doped DLC coatings (E = 84 ÷ 105 GPa and Poisson = 0.25 [21]) and the AISI316L counterbody (E = 190 GPa and Poisson = 0.265), and neglecting surface roughness influence, the Hertzian contact stresses ranged from 420 to 465 MPa, while the contact radius was approximately 32–33 um. Modified simulated body fluid (SBF) alone and with the addition of bovine serum albumin (BSA) were used as lubricants. Prior to this, the tests samples were incubated in fresh SBF or a BSA solution for 1 h at 37 °C. SBF was adjusted to the desired pH 7.4 using 1M HCl. The concentration of albumin in the simulated body fluid was 50 g/L (concentration of albumin in human blood plasma). Additional tests were carried out under dry friction condition in ambient air, at 24 °C, and in a relative humidity of 30–40%. For each type of coating the test was repeated twice and the results were averaged.

After the tribological tests, the resulting wear tracks were analyzed using the G200 nanoindenter (KLA Corporation, Milpitas, CA, USA). The wear profiles were registered by means of a diamond conical indenter with tip radius of 1 μm and an apex angle of 90° and specific wear rate was calculated. For each wear track four profiles were recorded and the results were averaged. The obtained wear tracks and wear scars were also investigated using the optical microscope MA200 (NIKON, Tokio, Japan). Based on the measurement of the diameter of the circle inscribed in the wear scar, the ball wear coefficient was determined based on the formula defining the volume of the spherical cap. The chemical structure analysis after the tribological tests in SBF and BSA was performed using an FTIR Nicolet iN10 microscope (Thermo Scientific, Waltham, MA, USA). The spectrum in the 4000–600 cm^−1^ range was recorded both from the wear track on randomly selected surface areas of 30 µm × 30 µm and from the undamaged part of the coating from 100 µm × 100 µm areas. The measurement was performed in reflection mode at a resolution of 2 cm^−1^, and the number of scans was 256. A high-sensitivity, nitrogen-cooled MCT detector was used for the tests. The changes in the chemical structure of a-C:H:SiO_x_ and a-C:H:Si coatings were determined by a Raman spectroscopy technique. The Raman spectra were recorded at room temperature using the inVia Raman Microscope (Renishaw, Gloucestershire, UK) with a 532 nm laser wavelength and a power of 0.3 mW. In order to detect the potential signal related to the presence of BSA and/or SBF on the sample, surface tests were carried out in the extended spectral range of 100–2000 cm^−1^. Chemical structure analyses of the coatings were performed by deconvolution of the Raman spectra (in the region of ~1000–2000 cm^−1^) with PeakFit (v4.12, Seasolve, San Jose, CA, USA) software using the Gaussian function. Subsequently, the G-peak position and the I(D)/I(G) ratio were determined. In each test the unmodified DLC coatings were investigated for comparison.

## 3. Results and Discussion

The samples were divided into two sets depending on the silicon precursor used for the deposition. In Table 1 the basic process parameters complemented by the concentrations of silicon for particular process parameters measured using X-ray photoelectron spectroscopy (XPS) [20,33] are presented. Our previous papers related to DLC coatings doped with silicon using HMDSO revealed that high negative self-bias potentials during the deposition improve the quality of the silicon-incorporated coatings [22]. Coatings deposited under the negative self-bias potential of 800 V were more uniform and were free from SiO_x_ inclusions with more stable tribological properties measured under dry friction conditions, as compared to those synthesized using the negative self-bias of 600 V [20,32]. Similarly, in the case of the studied coatings, they were uniform and homogeneous with no signs of cracks, delamination or other defects. The deposition time of each coating was optimized to a final thickness of 1 µm. As it is visible the lowest deposition rate is characteristic for pure DLC manufactured with use of CH_4_. Both silicon precursors increased the deposition rate, wherein for TMS its value was almost twice compared to HMDSO. Similar trend may be observed in the case of the silicon concentration. For TMS, despite practically the same flow ratio of the carbon precursor, the resulting concentrations of Si in TMS1 and TMS2 coatings were noticeably higher compared to HMDSO1 and HMDSO2 (ten and two times higher, respectively). The energy of dissociation of Si-O-Si bond is higher in comparison to Si-C. Therefore, as the result of dissociation of HMDSO many silicon atoms are still bonded to oxygen in the form of Si-O-Si bonds [34]. They are either incorporated into the deposited coating or more likely evacuated by the pumping unit. The remaining CH_3_ groups either dissociate to CH_2_ or remain intact and in both forms take part in the process of synthesis. In the case of oxygen free TMS the probability of dissociation of Si-C bonds is much higher and therefore, higher deposition rate and more incorporated silicon were registered in the case of TMS coatings.

### 3.1. Frictional Behavior

Figure 1 presents averaged values of the coefficient of friction of DLC and silicon-incorporated DLC coatings after the tests in ambient air, simulated body fluid, and bovine serum albumin, whereas Figure 2 presents graphs of friction coefficient evolution for each test. The average values of the COF for each test were calculated excluding running-in-period fluctuations of friction force (initial 35 m).

A positive effect of silicon admixture is visible in the case of all silicon-incorporated carbon coatings tested in ambient air conditions. The coatings synthesized using HMDSO show typical behavior, that means a very low value of the friction coefficient (0.05) for an HMDSO1 sample with a low concentration of silicon and oxygen [32]. Further increase in the concentration of silicon and oxygen increases the value of the COF (0.1) but it is still 40% lower compared to DLC. The COF evolutions along the test for each coating were rather stable and did not show any abrupt changes (Figure 2a), a finding that is in agreement with our earlier study [32]. An opposite trend for the friction coefficient may be observed for TMS1 and TMS2 samples, respectively, after 25% of the whole test (Figure 2b). The COF value of sample TMS1 linearly decreases, whereas for sample TMS2 it linearly increases to reach a very similar value to the former one at the end of the test. Nevertheless, the lower value of the coefficient of friction was registered for the coating denoted as TMS2 (0.11) which is similar to the one for coating HMDSO2 and still lower in comparison to the unmodified DLC. Note that both coatings contain similar concentrations of Si. 

The much lower values for the coefficient of friction of silicon incorporated carbon coatings are related to the formation of SiO-(H) functional groups on the surface or the formation of a thin film of SiO_2_ acting as a solid lubricant [32]. The positive influence of oxygen is clearly visible for the coating denoted as HMDSO1. However, it should be noted that here we do not have a reference for a sample with the same concentration of silicon deposited using TMS.

Simulated body fluid and bovine serum albumin environments create more demanding conditions for carbon-based coatings mostly due to their commonly known sensitivity to humid environments [35]. In Figure 2c,d, the evolution of the friction coefficient of all types of coatings tested in SBF is presented. At first glance a lack of COF stability is visible for the DLC coating. The COF, despite its slightly lower value compared to air atmosphere, shows fluctuations during the whole test, which was also noted by Hang et al. [26]. In the case of silicon incorporated coatings, again, the clearly positive effect of oxygen may be visible. Samples synthesized using HMDSO show lower COF values for both concentrations of silicon and oxygen (Figure 2c). In the case of the HMDSO1 sample, the coefficient of friction gradually decreases to reach a value around 0.11 at the end of the test, whereas the COF of the HMDSO2 sample increases, reaching a final value around 0.13. Coatings deposited with use of TMS for half of the total test duration show similar behavior to bare DLC (Figure 2d). Next, the COF values of the TMS1 and TMS2 samples gradually increase to 0.2 and 0.18, respectively, whereas the COF value for DLC stabilizes at around 0.16. In an aqueous environment, dangling bonds termination and Van der Waals force interactions between the tribopair and H_2_O molecules appear to be the main factors determining the friction force. In the case of the oxygen-free coatings, namely DLC and TMS, these interactions appear to be much stronger, thus increasing the friction coefficient. Coatings synthesized from the HMDSO/CH_4_ gas mixture contain silicon primarily bonded to oxygen. These bonds show high chemical polarity leading to high wettability. Since both, the sample and the counterbody materials are hydrophilic (contact angle of polished AISI 316L steel is usually around 71 deg), the friction force in this case may be dominated by the interactions between H_2_O molecules attached to both surfaces cooperating in a boundary friction regime. Note that silicon-incorporated oxygen free DLC coatings also present hydrophilic properties [36]. The contact angle of the tested TMS1 and TMS2 coatings was equal to 78.8 ± 0.39 and 76.5 ± 0.34, respectively (results not yet published). Although both types of analyzed coatings are directly exposed to oxygen during the friction test, FTIR results indicate their different chemical structure (discussed later). Moreover, the surface functional groups providing the hydrophilic properties of TMS coatings may be easily removed as the results of the tribological process.

The addition of proteins to SBF resulted in the opposite effect. The bovine serum albumin environment appears to be favorable for coatings synthesized using TMS (Figure 2f). The unmodified DLC coating presented a gradually increasing trend of the coefficient of friction up to ca. 4000 revolutions. After reaching a value of 0.27 a sudden drop down to 0.24 was observed with further stabilization at this value until the end of the test, this finding is in agreement with other literature reports [37]. In the case of both Si-incorporated coatings deposited using HMDSO, a decrease of the friction coefficient was registered (Figure 1). Nevertheless, the coating denoted as HMDSO1 presented a slightly lower, stable and uniform trend of the COF (ca. 0.21) as compared to HMDSO2 (Figure 2e). Moreover, at the end of the test the coefficient of friction for both the DLC and HMDSO2 coatings remained at the same level, close to 0.24. Noticeably lower values of the coefficient of friction have shown coatings deposited with the use of TMS. Since the very beginning of the test both coatings presented a low and stable COF value, whereas the value for the coating with a lower concentration of Si was noticeably lower, reaching a value of ca. 0.14. Along with the test progress, both coatings show a slightly increasing trend of the coefficient of friction, nevertheless their mutual relation is kept. Biomacromolecules adsorbing on the surface of the coatings influence their friction and wear behavior, and its analysis is twofold. Firstly, the attached proteins create contact sites between the coating and the counterbody, which may lead to the increased value of the coefficient of friction. On the other hand, the attached proteins may act as a protective agent and decrease the wear rate of the coating and the counterbody. In terms of the COF, the first hypothesis appears to be valid for DLC coatings which are known to be sensitive to a protein-containing environment [26,31]. Noticeably different behavior can be seen between Si-incorporated DLC coatings synthesized using an oxygen-containing and oxygen-free precursor. For both types of coatings an increase in silicon concentration negatively affects the value of the coefficient of friction, nevertheless its value is still lower in comparison to the DLC. However, coatings rich in oxygen (denoted as HMDSO1 and HMDSO2) appear to be more sensitive to the sliding environment which is reflected in higher values of the COF. Note that both the HMDSO2 and TMS1 coatings contain the same concentration of silicon. Lower values of the coefficient of friction characteristic for TMS coatings indicate that they are less prone to attach the proteins and create bridges between the sample and the counterbody (discussed later).

### 3.2. Wear Rate Analysis

Figure 3 and Figure 4 present images of wear tracks and wear scars on the coatings and counterbodies, respectively, after ball-on-disc tests performed in selected environments. The wear tracks after the sliding test in air indicate their abrasive nature with traces of the third body effect, especially in the case of the HMDSO2 and TMS coatings. In the periphery of the wear tracks a noticeable amount of the counterbody material is visible, this finding was also confirmed in our earlier paper [32]. For tests in both liquid solutions, SBF and BSA, dry residues of environmental products are clearly noticed, making the analysis barely possible. This is because the samples were left to dry after the test at room temperature (the compressed air was not used to not interfere with the wear track). Despite this, traces of low intensity abrasive wear are visible. The limited influence of the third body effect is caused by the liquid environment, enabling the continuous removal of worn counterbody material from the contact zone. Clearly, however, the wear track of the HMDSO1 coating indicates that the wear processes of this sample were definitely less severe, regardless the environment. 

The wear scars on the surface of the counterbody also indicate an abrasive character. Neither visible traces of the counterbody nor biological material were observed on the surface. Interestingly for the coating denoted HMDSO1 tested in air and the BSA environment, the wear scar is barely visible. Despite a few abrasive scratches on the surface, the wear scar does not show any distinguishable wear area. This indicates a very low wear rate of AISI316L counterbody cooperating with the HMDSO1 coating. The comparison between wear tracks and wear scars of the HMDSO and TMS samples tested in BSA confirms the protein adsorption hypothesis. Despite the fact of the higher COF of the HMDSO samples, their wear appears to be less severe due to the formation of a protective layer on the surface.

Based on the registered wear track profiles and wear scar diameters, specific wear rates of the friction pairs were calculated. The calculated wear rates are presented in Figure 5. The negative values of wear rates for the counterbodies were introduced conventionally for direct comparison of the wear rate of the coating and the counterbody. For tests conducted in the air (Figure 5a), the DLC coating shows one order of magnitude lower wear rate in comparison with other tested samples, although it does show the highest value of the coefficient of friction. As stated earlier, due to the lack of a visible wear area on the surface of the AISI316L counterbody sliding against the HMDSO1 coating, the wear rate could not be calculated, nevertheless its value was lower as compared to the DLC coating, which is in agreement with our earlier results [32]. In addition to the DLC and HMDSO1 coatings, the others showed significant wear rate both of the sample and the counterbody. The negative influence of increasing the concentration of silicon on the COF value is visible in the case of coatings produced with use of both silicon precursors.

Both SBF and BSA environments show superior performance of HMDSO coatings, wherein a higher concentration of Si in the HMDSO2 coating noticeably increased the wear rate of the counterbody, which still is lower as compared to both TMS coatings (Figure 5b,c). Similar to the test in air, for the test in BSA it was impossible to determine the wear rate of the counterbody after the test with the HMDSO1 layer (only a few scratches are visible on the surface indicating negligible wear rate). Coatings deposited using an oxygen-free precursor during tests in SBF and BSA environments show an opposite effect. Namely, in SBF the wear rates of the TMS1 and TMS2 coatings increased with the growing concentration of silicon, while the one for the counterbodies decreased. In BSA an increase in the concentration of silicon results in both decreased wear rate of the sample and the counterbody.

### 3.3. Chemical Characterization

The comparative FTIR spectra of the as deposited HMDSO2 and TMS1 coatings (both with similar silicon concentrations) are presented in Figure 6, whereas the location of the characteristic absorption bands is given in Table 2. Spectra obtained from both types of coatings are characterized by a low number of C=O and almost no C=C bonds. The TMS coating shows more pronounced Si-C and Si-O-C peaks, whereas for HMDSO the adsorption bands belong to Si-O bonds. In both cases this is related to the chemical nature of the precursors from which these coatings were obtained.

Figure 7 shows the FTIR spectra of silicon-incorporated DLC coatings (HMDSO2, TMS1) obtained from the wear tracks after the tests in the SBF and BSA environments. The spectra of the tested coatings show differences only in the range of 1800–800 cm^−1^, therefore this range of wavenumbers is presented. The location of the bands and their membership to specific types of bonds are additionally summarized in Table 3.

Coatings synthesized using TMS and tested in SBF solution, besides the typical bands for silicon-incorporated DLC films, are also characterized by the presence of C=O bonds at 1750, 1680, 1305 cm^−1^. An additional peak originating from the C–O–C bonds was also observed. The appearance of maxima derived from sp^2^ hybridized carbon (C=C) bonds is noteworthy. These bonds were not observed in the case of the HMDSO coatings. Moreover, for the TMS coatings two very pronounced sharp peaks originating from Si-O bonds (around 1057 and 1038 cm^−1^) may indicate a friction induced formation of SiO_x_ layer or inclusions. Both may be the reason for the accelerated wear of the sample and the counterbody. Distinct signs of abrasive wear and the third body effect registered in wear tracks and wear scars of cooperating elements together with the one order of magnitude higher wear rate of the counterbody seem to confirm this hypothesis. The spectra for the HMDSO coatings are much poorer. The bands belonging to Si-CH_3_, Si-O, Si-O-C and Si-C bonds are dominant. Other bands are characteristic of the DLC layers. Nevertheless, the HMDSO coatings perform much better in the SBF environment compared to TMS, showing a low coefficient of friction and a noticeably lower wear rate. The FTIR results indicate neither graphitization nor formation of SiO_x_, however on the surface of as deposited HMDSO coatings Si-OH functional groups were found and these are also considered to have a positive effect on reducing the coefficient of friction.

In the case of the wear tracks of samples tested in a BSA environment, differences in the course of the IR spectra of TMS and HMDSO coatings are also visible. A set of distinctive bands characteristic for albumin derived from N-H, C-N and C-C-N bonds were registered for HMDSO coatings. This confirms the high values of the coefficient of friction in the BSA environment, caused by the high surface adhesion of proteins. On the other hand, the low wear rate of the HMDSO coatings tested in BSA proves the protective properties of protein film. As can be seen in Table 3 and Figure 7, the TMS coatings have significantly lower intensities of these maxima. Furthermore, in the IR spectrum for the TMS coatings, the band characteristic for the C-N bond (around 1299 cm^−1^) was not identified. This means that the protein adheres much better to the HMDSO coatings than the TMS. It is also confirmed by the low values of the coefficient of friction and the high wear rate of the TMS coatings.

The Raman spectra of the analyzed coatings were typical for diamond-like carbon films with a broad asymmetric peak in the range 1400–1700 cm^−1^, which is in agreement with our previous results published elsewhere [20,33]. Results of the Raman spectroscopy test are presented in Figure 8. In the process of deconvolution of the spectra into D and G bands, the I(D)/I(G) intensity ratio and the position of the G band were determined. The I(D)/I(G) ratio refers to the content of sp2 hybridized carbon (relative to sp^3^ hybridization)-which means that the higher the I(D)/I(G) ratio is, the higher the content of sp^2^ hybridized carbon in the structure is, whereas changes in the position of the G band in the spectra of DLC coatings may result from changes in both the stress level and the content of sp^2^ hybridized carbon regions [45]. The analysis of the evolution of this parameter completes the observation of changes in the I(D)/I(G) ratio.

Figure 8a,b presents the I(D)/I(G) ratio and the position of the G band for coatings tested in the air atmosphere. As can be seen, the as-deposited DLC coating is characterized by the highest I(D)/I(G) ratio (Figure 8a), which significantly decreases in the case of silicon-incorporated coatings. It is related to the decreasing content of sp^2^ hybridized carbon bonds due to the occurrence of silicon, which, as a tetravalent element, promotes the formation of sp^3^ hybridized orbitals. This observation is consistent with other literature reports [20,46]. For the HMDSO1 coating, the I(D)/I(G) ratio is the highest among the doped coatings, which is logical as they have to the lowest silicon concentration equal to 0.45. It is worth noting, however, that the TMS2 coating with the highest silicon concentration (10 at.%) does not show the lowest I(D)/I(G) ratio (which characterizes the HMDSO2 coating). Therefore, it can be concluded that the increase in the concentration of sp^3^ hybridized carbon in the structure of doped DLC coatings is not determined only by the concentration of silicon in the amorphous matrix of the DLC. Besides the chemical composition of the precursor (which, apart from Si, also contains hydrocarbon groups, and in the case of HMDSO-oxygen), its chemical structure is also important, influencing the dissociation character under given glow discharge conditions.

The literature reports show that doping DLC coatings with silicon contributes to the improvement of their thermal stability [47,48]. Moreover, in the case of dry friction conditions, the addition of silicon inhibits the thermally induced sp^3^ to sp^2^ transformation [32]. Similar to those observations, both HMDSO and TMS coatings have shown smaller increase in the I(D)/I(G) ratio in the wear tracks, which proves their slight graphitization compared to the undoped coating. Among the doped coatings, the highest increase in the I(D)/I(G) ratio after tribological tests was observed for the coating with the lowest Si concentration (HMDSO1). Taking into account the location of the G band for the as-deposited coatings, in the case of silicon-doped coatings, a shift towards lower wavenumbers is observed, which may indicate a reduction of carbon areas with sp^2^ hybridization (it is consistent with the evolution of the I(D)/I(G) ratio). It is also commonly known that incorporation of silicon into the DLC matrix decreases the level of internal stress, which may also contribute to the shift of the G band towards lower wavenumbers [21]. The analysis of changes in the position of the G band after the tribological tests confirms the conclusions drawn based on the changes in the I(D)/I(G) intensity ratio. Namely, the observed shift of the G band towards higher wavenumbers may indicate an increase in the degree of graphitization of the coatings.

Figure 8c,d presents values of the I(D)/I(G) ratio as well as the position of the G band of the coatings subjected to the tribological tests in SBF solution. Note that similar to the wear track analysis, the obtained test results concern the surfaces with remains of SBF solution. A high light absorption coefficient characteristic for carbon materials allows the laser to penetrate the surface of the DLC coatings to a depth in the range of approximately 10~100 nm. Due to this fact, it can be assumed that the obtained results are not determined only by the presence of the C-H groups of the SBF solution, but they represent the averaged value (along with the matrix parameters) from the depth range indicated above. Depending on the type of coating, the change of these parameters in relation to as deposited coatings show a different tendency. This may indicate a different degree of adsorption of the SBF solution to their surface, related to the differences in their chemical composition and thus a different surface energy (which affects the adsorption as discussed earlier).

At first glance all analyzed coatings do not show graphitization after the tribological tests in SBF solution, or they show it to a small extent. This may indicate that SBF solution acts as a lubricant. However, considering the general increase in the COF of Si-DLC coatings and the barely noticeable decrease in the case of DLC tested in SBF solution, it more likely ensures a very good heat transfer and thus, inhibits the graphitization of the coating in the wear track. It is worth noting that the values of the I(D)/I(G) and G-pos. parameters for DLC coatings before and after the tribological test in the SBF solution do not change, unlike the Si-doped coatings. Here, TMS coatings show a lower I(D)/I(G) ratio, whereas for HMDSO coatings one increases (the G-band position shows similar tendency). Note that during the tests in SBF the HMDSO coatings performed better, showing lower coefficient of friction values and a lower wear rate as compared with the DLC and TMS layers. Since the chemical structure of TMS coatings could not change in the opposite direction (i.e., the increase in the concentration of sp^3^ hybridized carbon bonds after the tribological test) it proves, that apart from Si admixture, the composition and chemical structure of the precursor used to produce the coating are also important in terms of the analysis of the surface properties of layers tested in SBF solution. The differences in values of the I(D)/I(G) ratio and positions of the G-band between as deposited coatings and the area contacted with the medium but not taking part in the tribological process (grey bars on the graphs) are also noteworthy.

The Raman spectroscopy results for the coatings tested in the BSA environment are presented in Figure 8e,f. Similar to the tests in SFB, the results did not reveal the bands characteristic for this solution and the values of the I(D)/I(G) ratio and the position of the G band just after contact with BSA are different from these as deposited, excluding DLC. The surface of the DLC coating shows the same I(D)/I(G) values, thus it seems that this surface is not as good an adsorbent for the BSA solution as it is for SBF and that numerous CH_3_ and CH_2_ groups did not affect the measurement result. A surprisingly large (and the highest among the tested samples) increase in the value of the I(D)/I(G) ratio was observed for the HMDSO1 coating. It has the lowest silicon content, therefore, one would expect a similar I(D)/I(G) ratio as for DLC. On the other hand, this coating definitely contains more oxygen (9.3%) than silicon (Si/O ratio = 0.05), which could suggest the crucial role of this element in the BSA adsorption process, as confirmed by FTIR.

After the tribological tests in BSA, all coatings have shown an increase in the I(D)/I(G) ratio (i.e., graphitization), although this is still to a lesser extent than in the case of the tests in the air atmosphere. Here an undisputable winner is the coating denoted as HMDSO1, showing the least visible increase in the I(D)/I(G) ratio and a lack of shift of the G band towards higher wavenumbers. This observation confirms our earlier hypothesis that coatings deposited with use of HMDSO are prone to attach biomacromolecules which on the one hand increase the coefficient of friction due to the formation of bridges between the adsorbed proteins and on the other hand protect the surface of the sample and the counterbody against excessive wear or graphitization. The results of the tribological tests and FTIR spectroscopy are in agreement with this statement. The difference in the coatings’ behavior during the tests in SBF and BSA, besides the tendency to attach proteins, may result from a much poorer thermal conductivity of the sliding environment caused by the bovine serum albumin adsorbed on the cooperating surfaces. The highest increase in the I(D)/I(G) ratio was registered for the DLC coating. Similar observations were reported by Hang and Qi, examining the structure of the DLC coating after the tribological test in air, HSF (human serum fluid), and BSA [26].

### 3.4. Model of Interaction between BSA and Silicon-Incorporated DLC Films

Several studies have addressed the role of albumin on the tribological behavior of DLC coatings. There are reports stating that the biomacromolecules can adsorb on the surfaces of the articulating materials, and strongly influence their friction and wear behavior [27,28,29,30]. The effect of the aqueous environments, including simulated body fluid and human serum albumin solution, on the friction and wear of DLC coatings has been studied and discussed by Huang et al. [26]. The proposed mechanism of friction [26] indicate, that the excess of H_2_O molecules in the aqueous solution causes almost all dangling bonds to be terminated with C-H, C-O and C-OOH groups. Bovine serum albumin of a high molecule weight may form many contact sites between the DLC coating and the counterbody via forming hydrogen bonding and Van der Waals interaction between a tribopair and H_2_O molecules. Since an additional force must be applied to conquer them, the resulting coefficient of friction increases, what is attributed to the formation of contact sites between the sliding surfaces and denatured protein [26,31]. According to the authors the significant differences in the adsorbed amounts of protein onto the analyzed surfaces did not affect their tribological behavior.

In our tests of silicon-incorporated DLC coatings in SBF and BSA, lower COF values were obtained as compared to the air environment. The entry of proteins into the coating-counterbody contact zone will depend on the surface drag forces of the contacting bodies and the suspended material and the position of these proteins in relation to the central flow [49]. The schematic illustration of BSA–silicon-incorporated DLC coatings interaction is presented in Figure 9. Note that both the counterbody as well as the silicon-incorporated DLC coatings show hydrophilic properties (discussed earlier), therefore friction force resulted in the interaction between H_2_O molecules and the tribopair. Before the measurement, the TMS and HMDSO coatings were left in the BSA solution, so the proteins adhered to their surface in the native form. During the wear studies, the presence of all possible BSA forms should be considered, including native/folded proteins adsorbed to the surface, free molecules loosely suspended in the solution, denatured ones and larger proteins agglomerates.

Probably, the majority of proteins flow around the contact periphery. As a consequence, depending on the BSA concentration, many of its agglomerates could be created and will cover the silicon-incorporated DLC coatings as a thicker film. During the wear test, folded proteins are then dragged into the contact, thus causing changes in BSA conformation to denatured and sheared. The area where the BSA enters the contact was marked in Figure 9 and named the inlet zone. The closer the protein molecule is to the central flow line, the greater the probability for contact entry [50].

For the HMDSO coating, characteristic peaks from peptide bond vibration were observed by FTIR spectroscopy, that confirms the presence of folded BSA. In the case of the TMS coatings, we observed a lower maxima of peaks intensities and no vibration of the C-N bond, which may indicate the denaturation of proteins during wear tests. Due to the increase in local temperature, trace amounts of denatured protein may be present in the wear tracks for both the TMS and HMDSO coatings. 

Summing up, the role of BSA as an agent preventing surface degradation in the sliding system can be dual. Firstly, proteins can act as a lubricant, forming a complex adsorbed film including BSA together with metallic debris/ions. On the other hand, BSA can improve the stability of the passive film acting as a corrosion barrier.

## 4. Conclusions

Silicon-incorporated coatings with different contents of silicon were deposited using oxygen-containing (hexamethyldisiloxane) and oxygen-free (tetramethylsilane) precursors. Coatings denoted as HMDSO2 and TMS1 were characterized by the same concentration of silicon, except the HMDSO2 coating additionally contained oxygen originating from the precursor. The coatings were tested in air, SBF, and BSA environments using the ball-on-disc tribometer. In the air atmosphere both types of coatings showed low and stable values of the coefficient of friction with a slight advantage of coatings containing oxygen. In the aqueous environment, a-C:H:SiO_x_ coatings were characterized by noticeably lower COF values as compared to the oxygen-free layers. In the case of TMS coatings, the coefficient of friction was comparable to unmodified DLC, regardless of the concentration of silicon. In the liquid environment the wear rate of coatings with a low concentration of silicon and oxygen was much lower in comparison with DLC and TMS layers. Moreover, an excess of SiO_x_ on the surface of TMS coatings registered after the test in SBF possibly was the reason for the accelerated wear of the counterbody, showing signs of the abrasive wear with the third body effect. In the BSA environment, TMS coatings has lower friction coefficient values, which may be related to the lower attachment of proteins to the surface. The oxygen in HDMSO coatings negatively affects their COF due to its divalent nature, and thus greater ability to bind to protein macromolecules. However, this ensured a much lower wear rate of the samples and the counterbodies. In general Si-DLC coatings tested in SBF and BSA have shown a lower wear rate in comparison to the DLC, unlike in the test in air. The optimal silicon content and chemical composition of the precursor used for the synthesis of silicon-incorporated DLC coatings could be one of the key points for materials manufacturing in the context of biomedical applications. Our results indicate that the coatings with low concentrations of silicon, deposited with use of the HMDSO precursor appear to be better candidates.

## Figures and Tables

**Figure 1 materials-15-02082-f001:**
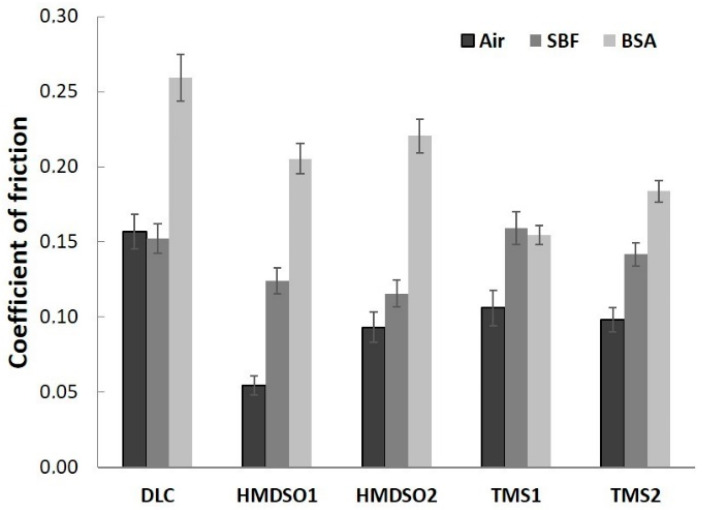
Comparison of the friction coefficient of silicon-incorporated DLC coatings in different environments: air, SBF, and BSA.

**Figure 2 materials-15-02082-f002:**
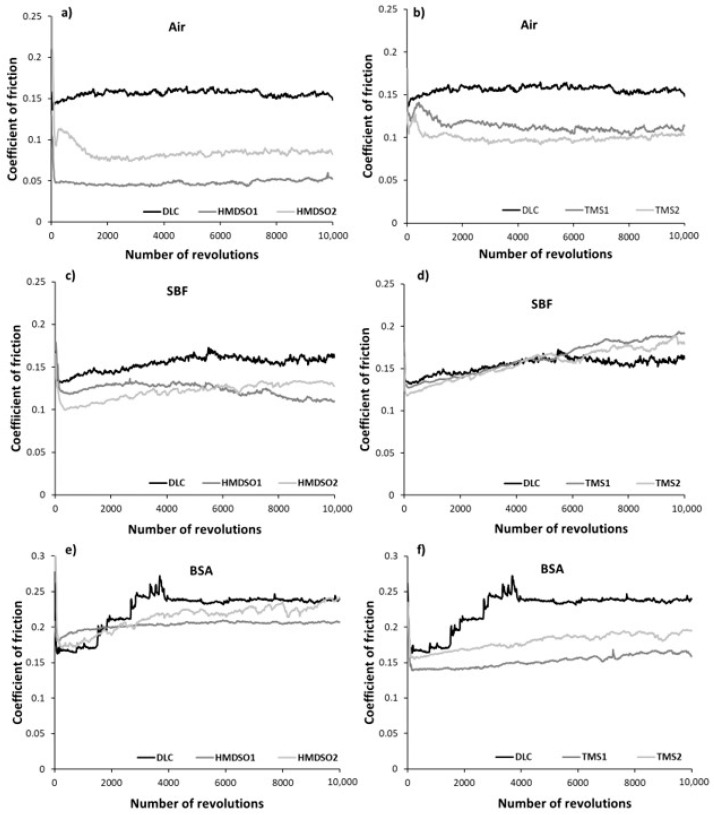
Friction coefficient evolution of DLC and silicon-incorporated DLC coatings synthesized using both HMDSO (left column) and TMS (right column) tested in (**a**,**b**) air, (**c**,**d**) SBF, and (**e**,**f**) BSA, respectively.

**Figure 3 materials-15-02082-f003:**
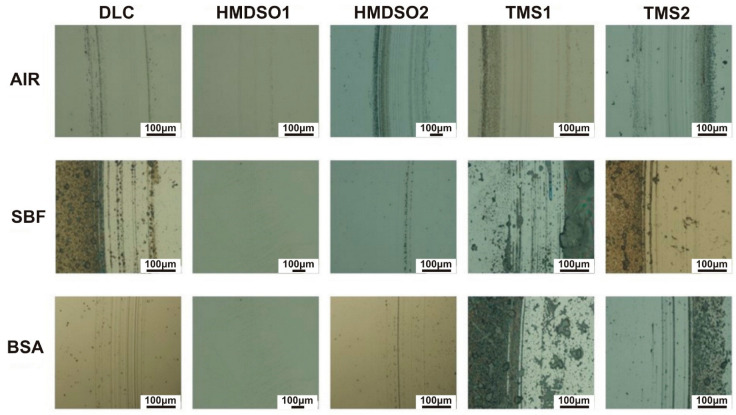
Wear tracks on the surface of DLC and silicon-incorporated DLC coatings tested in different environments. The scale bar length on all images is equal to 100 µm.

**Figure 4 materials-15-02082-f004:**
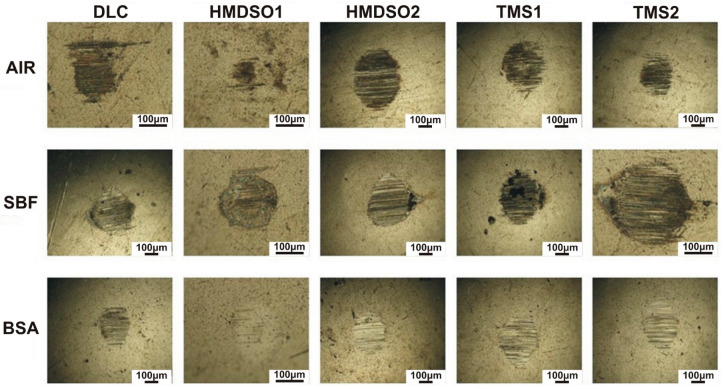
Wear scars on the AISI316L counterbody tested with DLC and silicon-incorporated DLC coatings in different environments. The scale bar length on all images is 100 µm.

**Figure 5 materials-15-02082-f005:**
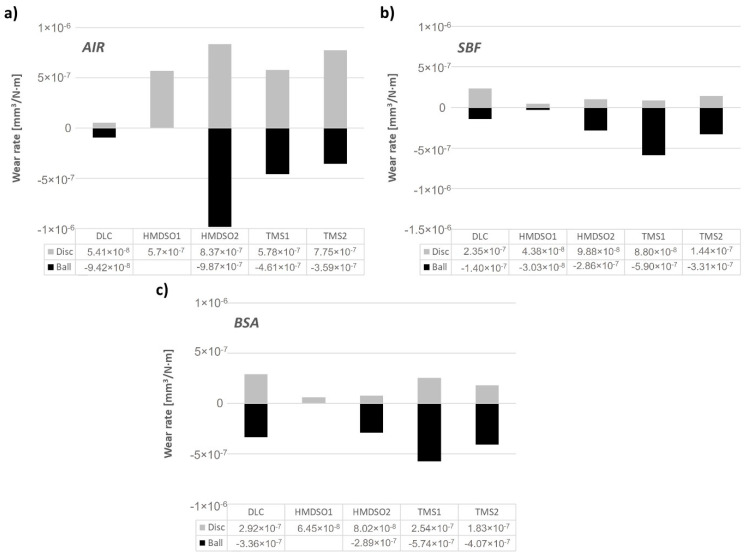
Wear rate of DLC and silicon-incorporated DLC coatings as well as the AISI316L counterbody sliding in (**a**) air, (**b**) SBF, and (**c**) BSA environment.

**Figure 6 materials-15-02082-f006:**
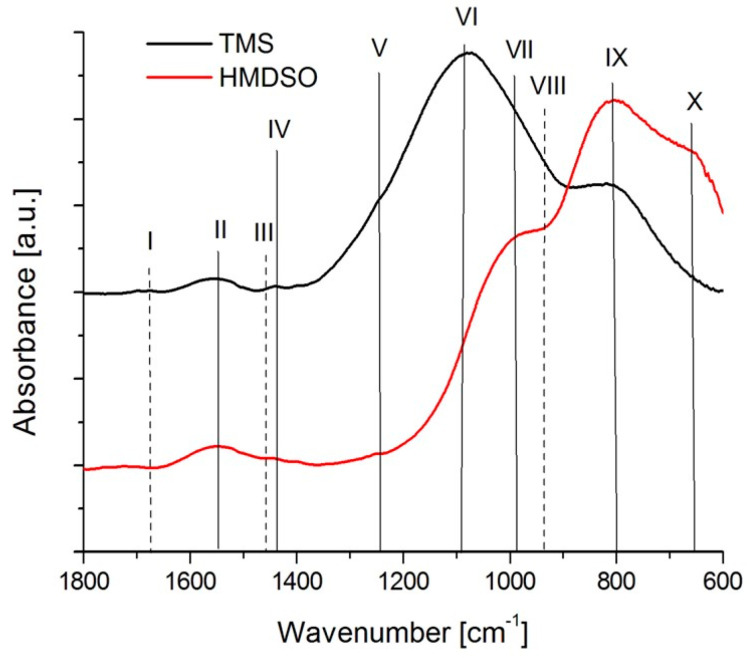
FTIR spectra of as deposited silicon-incorporated DLC coatings.

**Figure 7 materials-15-02082-f007:**
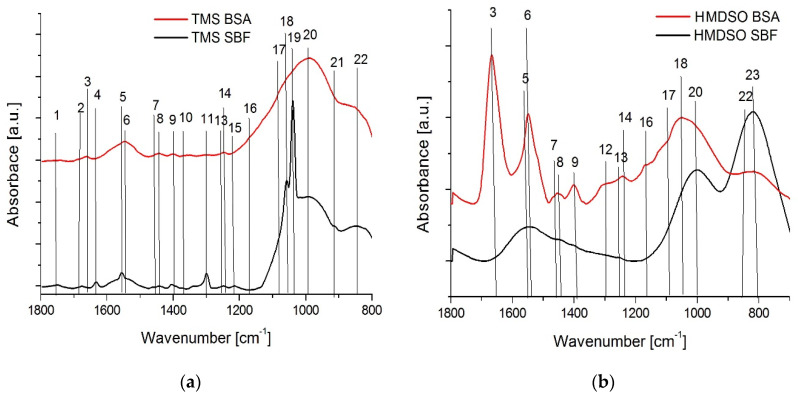
FTIR spectra of silicon-incorporated DLC coatings (**a**) deposited using TMS and (**b**) deposited using HMDSO, after ball-on-disc tests in SBF and BSA environments.

**Figure 8 materials-15-02082-f008:**
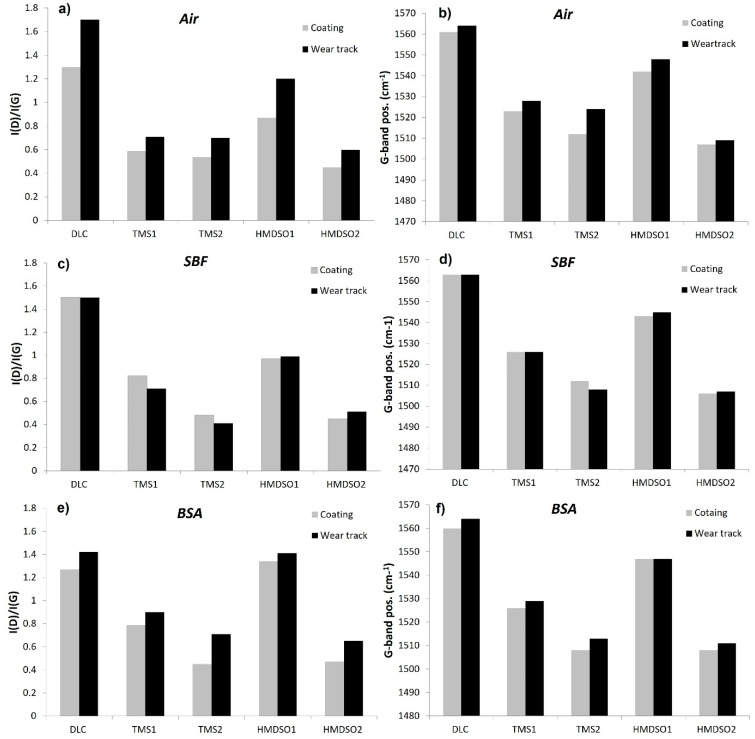
Changes in the ID/IG ratio (**left column**) and G-band position (**right column**) of the as-deposited coatings before and after the tribological tests in (**a**,**b**) air, (**c**,**d**) SBF, and (**e**,**f**) BSA environment.

**Figure 9 materials-15-02082-f009:**
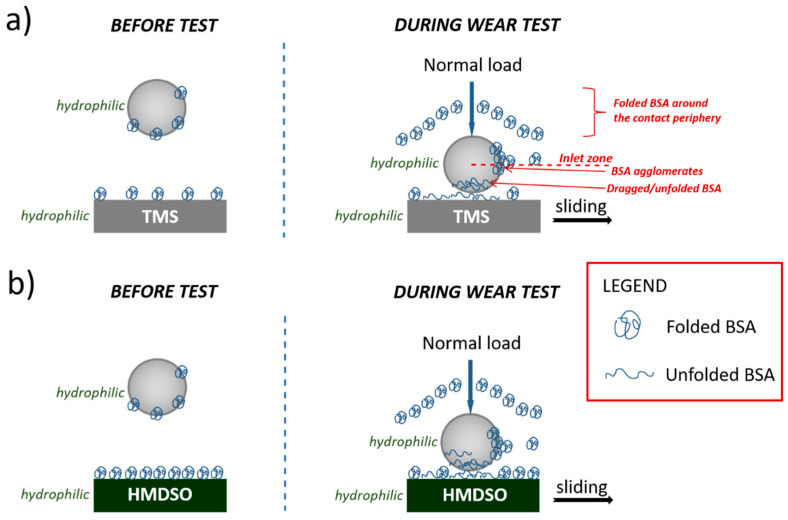
Model of interaction between BSA and silicon-incorporated DLC films synthesized using (**a**) TMS and (**b**) HDMSO.

**Table 1 materials-15-02082-t001:** Basic parameters of synthesis of 1 µm thick DLC and silicon-incorporated DLC layers.

Coating	Pressure [Pa]	Gas Mixture and Flow Ratio	Bias	Deposition Time	Si Content [at.%]	O Content [at.%]
DLC	20	CH4	−800	37	-	-
TMS1		CH4/TMS-18/4		17	5.0	-
TMS2		CH4/TMS-17/8		15	10.0	-
HMDSO1		CH4/HMDSO-18/3		28	0.45	9.3
HMDSO2		CH4/HMDSO-16/12		21	5.3	10.3

**Table 2 materials-15-02082-t002:** Characteristic IR absorption bands in the range 1800–600 cm^−1^ for as deposited silicon-incorporated DLC coatings.

No.	Wavenumber [cm^−1^]	Vibrating Mode [cm^−1^]	Ref.	TMS	HMDSO
I	1680	C=O (stretch)	[38]	+(weak)	-
II	1550	C=O (stretch)	[39]	+	+
III	1460	CH_3_ (deformation)	[38]	+	+
IV	1440	C-OH/Si-OH (bend)	[38,40,41]	+	-
V	1250	Si-CH_3_ (bend)	[41,42]	+	+
VI	1080	Si-O-C/Si-CH_2_-Si (stretch)	[43,44]	+(strong)	+(weak)
VII	998	Si-O(stretch)	[41,42,43]	+	+(strong)
VIII	940	Si-OH/Si-O (stretch)	[41,42]	+	+
IX	798	Si-C (stretch)	[41,42]	+(strong)	+
X	660	Si-O (bend)	[38,40,43]	+	-

**Table 3 materials-15-02082-t003:** Characteristic IR absorption bands in the range of 1800–700 cm^−1^ registered in the wear track of the analyzed silicon-incorporated DLC coatings.

No.	Wavenumber [cm^−1^]	Vibrating Mode [cm^−1^]	Ref.	TMS SBF	TMS BSA	HMDSO SBF	HMDSO BSA
1	1750	C=O (stretch)	[38]	+	-	-	-
2	1680	C=O (stretch)	[38]	+	+	-	-
3	1659	C=O, C-N (stretch) (amid I)	[38,40]	-	+	-	+(strong)
4	1630	C=C (stretch)	[38]	+	-	-	-
5	1550	C=O (stretch)	[39]	+(strong)	+	+	+
6	1540	N-H, C-N (deformation) (amid II)	[38,40]	-	+	-	+(strong)
7	1460	CH_3_ (deformation)	[38]	+	+	+	+
8	1440	C-OH/Si-OH (Bend)	[38,41,42]	+	+	+	+
9	1390	C-N (stretch) (amide III band)	[38,41]	-	+	-	+(strong)
10	1370	SiOCOCH_3_	[41,42]	+(weak)	+(weak)	-	-
11	1305	C=O (stretch)	[38]	+(strong)	-	-	-
12	1299	C-N (stretch) (amide III)	[38,40]	-	-	-	+
13	1250	Si-CH_3_ (bend)	[41,42]	+	+	+	+
14	1240	C-N (stretch)	[38,40]	-	+	-	+
15	1213	C-O-C (stretch)	[38]	+	-	-	-
16	1170	C-C-N (stretch)	[38]	-	+	-	+
17	1080	Si-O-C/R1-Si-O-Si-R2	[43,44]	+	+	+	+
18	1057	(stretch)	[41,42,43]	+(strong)	+	+	+
19	1038	Si-O/Si-CH_2_-Si (stretch)	[41,42,43]	+(strong)	+	-	-
20	1000	Si-O-Si (stretch)	[41,42,43]	+	+	+	+
21	920	Si-O(stretch)	[38]	+	-	-	-
22	857	CH=CH (deformation)	[44]	+	+	+	+
23	802	Si-O-C (stretch)	[41,42]	+	+	+	+

## Data Availability

Not applicable.

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
