# Peer review of "Tribological Characteristics of a-C:H:Si and a-C:H:SiOx Coatings Tested in Simulated Body Fluid and Protein Environment"

_materials, 2022, doi:10.3390/ma15062082_

Round 1

Reviewer 1 Report

The paper entitled “Tribological characteristics of a-C:H:Si and a-C:H:SiOx coatings tested in simulated body fluid and protein environ” aimed to present the effect of silicon and oxygen admixture on friction and wear behaviour of silicon-incorporated DLC coatings tested in SBF and BSA environments. The paper has scientific interest and originality in its technical content to merit publication. It presents some minor lack of explanation that can be improved. I am sending a copy of the manuscript (pdf) in which all the suggestions/corrections are highlighted.

Author Response

We would like to thank the reviewer for valuable and constructive suggestions to improve our manuscript. Below you will find our detailed responses to specific issues. Changes in the revised version of the manuscript and our responses are marked in yellow.

Why did the authors choose these contact conditions ...?
The authors must specify the mechanical properties ...

The choice of such contact conditions was indirectly related to the fact that there is no standard determining the range of loads that should be used during the test. On the other hand, the test parameters presented in the paper were selected with a view to ensuring the possibility of comparing the obtained results with the results of tribological tests carried out by other authors who used similar load ranges. The mechanical properties were added as suggested.

I can't understand why the authors did't plot for each lubricated conditions, all the curves on the same graph and they repeat the DLC curve?

We tried to organize the data in a way the reviewer suggests, however similar ranges of CoF values for both lubricating conditions make the graphs fuzzy and difficult to analyze. Therefore we decided to present the trends of CoF separately for each type of coating and lubricating conditions. The DLC curve is repeated for comparison. 

The authors should specify in Materials and Methods how they measured the wear of the spherical cup

We thank the reviewer for this suggestion. The proper information was added to the text.

Reviewer 2 Report

In this manuscript, the silicon-incorporated coatings with different content of silicon were deposited usinghexamethyldisiloxane and tetramethylsilane precursors. The friction properties and friction mechansim of coatings were investigated in detail in air, SBF and BSA environments using the ball-on-disc tribometer. Interaction between the coating and the working environment was analyzed. The a-C:H:SiOx coatings showed noticeably lower COF values as compared to the oxygen-free layers in aqueous environment. In BSA environment TMS coatings have lower values of friction coefficient due to the lower attachment of proteins to the surface. The work is important and would be of interest to the readers. Therefore, I recommend that this work can be accepted after a minor revision.

1, The microstructure of Si-incorporated DLC coatings should be characterized

2, The scale bar in Fig3 and Fig4 is not clear.

3, The corresponding Raman spectra in Fig.8 should be provided besides the ID/IG, G peak position.

4, The interpretation about the friction mechanism of Si-DLC in SBF and BSA environments at microscopic level need to be enhanced.

Author Response

We would like to thank the reviewer for valuable and constructive suggestions to improve our manuscript. Below you will find our detailed responses to specific issues. Changes in the revised version of the manuscript and our responses are marked in yellow.

The microstructure of Si-incorporated DLC coatings should be characterized.

We thank the reviewer for this comment. The analyzed carbon coatings are amorphous as is indicated in the title (a-C:H), so there is not much to add in terms of microstructure. As we state in the results section “Coatings deposited under the negative self-bias potential of 800V were more uniform and free from SiOx inclusions”. Moreover, we refer to our two other papers devoted to in-depth analysis of both types of coatings in terms of chemical structure and composition. In the case of a-C:H:SiOx coatings we added more appropriate reference. 

The scale bar in Fig3 and Fig4 is not clear.

We thank the reviewer for this comment. Indeed, the scale bars are not very clear as there are five pictures organized in one row, therefore in the figure caption we mention that the scale bar length on all images is equal to 100 µm.

The corresponding Raman spectra in Fig.8 should be provided besides the ID/IG, G peak position.

We thank the reviewer for this comment. We agree that the spectra should be provided. However, bearing in mind the current volume of the article and already rich graphic documentation we decided to add a reference to our other papers where we analyzed the Raman spectroscopy results for both types of coatings. We also added a proper comment in this regard.

The interpretation about the friction mechanism of Si-DLC in SBF and BSA environments at microscopic level need to be enhanced.

We thank the reviewer for this comment. We agree that the friction mechanism is not fully understood and explained. However, comprehensive and thorough analysis of friction mechanism in this case requires additional tests significantly beyond the scope of our work. First of all, a detailed surface chemical composition analysis should be performed with use of at least XPS technique, with special emphasis put on the wear track analysis. Secondly, 2D electrophoresis of the BSA residues from the wear track should be done and the results should be compared with those for native peptides. Without these results further analysis of the friction mechanism in our opinion would be based on guesses and hypotheses of low scientific level. The main aim of our paper was to present the differences in friction coefficient and wear rate of two types of carbon coatings modified with silicon, working in two different simulated body environments. The subject raised by the reviewer undoubtedly is very interesting and worth of further investigation, however, in the case of this paper, as we stated earlier, the volume of this work is already quite large. Surely we will focus our future work on this topic.

Reviewer 3 Report

The study deals with the DLC-Si coating. The application of DLC-Si in medicine is the main aim of the research, therefore the tribological tests were carried out in synthetic fluids simulating the human body fluids. This type of tests is an advantage of the present publication allowing a better understanding of wear mechanisms under these conditions, thus the results present an interest for scientific society.

There are only minor questions, comments and suggestions.

  1. The insertion of DLC coating in to human body requires special restrictions. HDMSO and TMS are the substances used in deposition process, if DLC-Si is not polluted during the high temperature deposition process by dangerous compounds formed due to decomposition of HDMSO and TMS?
  2. Wear product influences the quality of implant. The results show a higher wear on the AISI 316L stainless steel balls, probably the balls coated with DLC-Si or other coatings should be used?
  3. How important can be fatigue strength of coated samples (cyclic loading), in addition to abrasive wear, particularly for DLC film with high internal stress? Probably, the adhesive tests should be done also after the tribological tests in synthetic fluids as well?

Author Response

We would like to thank the reviewer for valuable and constructive comments. Below you will find our detailed responses to your questions.

The insertion of DLC coating in to human body requires special restrictions. HDMSO and TMS are the substances used in deposition process, if DLC-Si is not polluted during the high temperature deposition process by dangerous compounds formed due to decomposition of HDMSO and TMS?

Indeed, the implantation of any material into the human body requires special restrictions. In the case of DLC coatings produced by PA CVD method they are mostly manufactured from methane atmosphere. HMDS and TMS are organosilicon compounds containing, besides carbon and hydrogen, also silicon (TMS) or silicon and oxygen (HMDSO). During the deposition of the coating they dissociate in glow discharge producing variety of functional groups based on carbon, hydrogen, silicon and oxygen, which subsequently are deposited on the modified surface. None of those groups is harmful to the human body, therefore there is no such a risk.

Wear product influences the quality of implant. The results show a higher wear on the AISI 316L stainless steel balls, probably the balls coated with DLC-Si or other coatings should be used?

We thank the reviewer for this comment. We agree that the counterbody can be modified with use of the same coating material or can be replaced by ceramics. Different material associations for biomedical friction nodes can be found in the literature. Our earlier experience also indicates that mutual frictional interaction of carbon coatings modified with silicon and oxygen with steel or ceramic  counterbodies may be completely different. That is why we started our tests using the standard metallic biomaterial used by us as a reference.

How important can be fatigue strength of coated samples (cyclic loading), in addition to abrasive wear, particularly for DLC film with high internal stress? Probably, the adhesive tests should be done also after the tribological tests in synthetic fluids as well?

We thank the reviewer for this remark and advice. Undoubtedly the issue of fatigue strength in the case of coatings for medical implants is the critical parameter. Especially in the case of coatings with high residual stress. In the case of the analyzed coatings we deal with a high stress by incorporation of silicon. Our earlier results show that the level of compressive stress can be decreased down to less than 1 GPa. Nevertheless, we do fatigue strength tests of our coatings. The test are being conducted with use of hip simulator with special emphasis put on the adhesion, cracking and delamination of the analyzed film. Our paper in the present form is the first step, aimed on presenting  the differences in friction coefficient and wear rate of two types of carbon coatings modified with silicon, working in two different simulated body environments. The next step surely will be focused on further research of coatings showing the best performance in the preliminary pin-on-disc tests including the fatigue strength and adhesion analysis.